# An off-lattice discrete model to characterise filamentous yeast colony morphology

Kai Li[1], J. Edward F. Green[1], Hayden Tronnolone[2], Alexander K. Y. Tam[3], Andrew J. Black[1], Jennifer M. Gardner[4], Joanna F. Sundstrom[4], Vladimir Jiranek[4,5]*, Benjamin J. Binder[1]

**1** School of Computer and Mathematical Sciences, University of Adelaide, Adelaide SA, Australia, **2** College of Science and Engineering, Flinders University, Adelaide SA, Australia, **3** UniSA STEM, The University of South Australia, Mawson Lakes SA, Australia, **4** Discipline of Wine Science, Waite Campus, University of Adelaide, Urrbrae SA, Australia, **5** School of Biological Sciences, The University of Southampton, Southampton, United Kingdom

* V.Jiranek@soton.ac.uk

**Data Availability Statement:** All the data, code and materials can be found on GitHub: https://github.com/kaili2019/Li2024-off-lattice.

## Abstract

We combine an off-lattice agent-based mathematical model and experimentation to explore filamentous growth of a yeast colony. Under environmental stress, *Saccharomyces cerevisiae* yeast cells can transition from a bipolar (sated) to unipolar (pseudohyphal) budding mechanism, where cells elongate and bud end-to-end. This budding asymmetry yields spatially non-uniform growth, where filaments extend away from the colony centre, foraging for food. We use approximate Bayesian computation to quantify how individual cell budding mechanisms give rise to spatial patterns observed in experiments. We apply this method of parameter inference to experimental images of colonies of two strains of *S. cerevisiae*, in low and high nutrient environments. The colony size at the transition from sated to pseudohyphal growth, and a forking mechanism for pseudohyphal cell proliferation are the key features driving colony morphology. Simulations run with the most likely inferred parameters produce colony morphologies that closely resemble experimental results.

## Author summary

Yeasts are one of the most-studied organisms in biology due to their widespread use in food and beverage production and their role as a model organism in biomedical research. In this work, we combine mathematical modelling with experimentation to better understand the growth mechanisms of a yeast colony. Typically, yeast cells are spherical, but under environmental stress new daughter cells can become elongated. This change in the cell shape creates non-uniform growth in the colony, where filaments consisting of chains of elongated cells extend away from the colony centre, foraging for food. We use computer simulations to understand how individual cell shapes and reproduction mechanisms produce different colony patterns observed in experiments. The colony size at the sated-to-pseudohyphal transition and a forking in the cell proliferation mechanism were the key features that determine the shape and structure of a colony. We showed that by using the

**Funding:** KL acknowledges funding from the Australian Government through a Research Training Programme Scholarship. BJB, JEFG, AKYT, and VJ acknowledge funding from the Australian Research Council (Grant numbers DP230100406). In addition, AKYT would like to acknowledge funding from the Australian Research Council (DE240100897). The funders had no role in study design, data collection and analysis, decision to publish, or preparation of the manuscript.

**Competing interests:** The authors have declared that no competing interests exist.

most likely set of parameters, our computer simulation is able to generate colonies resembling those seen in experiments.

## Introduction and background

In this paper, we combine agent-based mathematical modelling, experiments, and statistical inference to investigate how cellular behaviour drives colony-scale patterns in filamentous yeast colonies. Yeasts are common single-cell fungal organisms, with over 1,500 recognised species. One of these species, *Saccharomyces cerevisiae*, was the first eukaryote to have its genome fully sequenced [1], and has since been one of the most widely studied model organisms in biology [2]. In addition to their use in research, yeasts positively impact everyday life through the production of baked goods, alcoholic beverages [3], medicines [4], and biofuels [5]. Other yeasts, for example *Candida albicans*, have a major negative impact of causing pathogenic infections in humans [6]. These impacts are strongly influenced by the morphology of individual cells or colonies. Therefore, understanding the fundamental mechanisms of yeast growth can benefit food production, biotechnology, and disease research. Depending on the strain and experimental conditions, colonies of lab-grown yeasts exhibit vastly different modes of growth.

We focus on small filamentous yeast colonies of two different strains relevant to wine research, AWRI 796 and Simi White [7]. Example colonies are shown in Fig 1. Although these two strains are the same organism, they produce visually different colony morphologies, which we seek to explain through our modelling. These filamentous colonies feature a central region of Eden-like [8] circular growth, and an outer region of non-uniform filamentous growth. In 1967, [9] showed that for bacteria grown in low-nutrient environments, only the outer region of the colony has access to nutrients, resulting in a constant increase in radius rather than an exponential increase in area. This phenomenon is known as diffusion-limited growth (DLG). Agent-based models for bacterial colonies have since shown that DLG can produce non-uniform patterns [10, 11] reminiscent of diffusion-limited aggregation [12]. Yeast colony development was assumed to be analogous to bacterial growth governed by DLG, but experimental work by [13] provided an alternative explanation for the non-uniform pattern. Their work showed that yeasts can actively respond to environmental stressors such as low nutrition by switching to a pseudohyphal growth mode [14]. This pseudohyphal growth involves cells

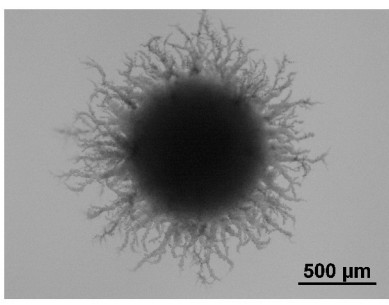
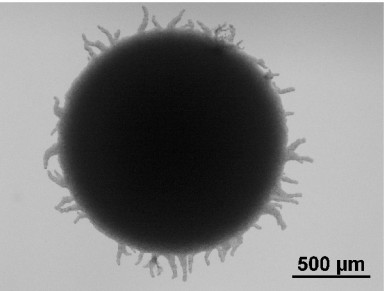
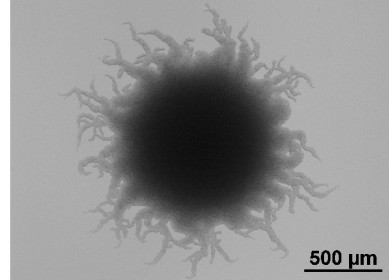

(a) AWRI 796 50 μM          (b) AWRI 796 500 μM          (c) Simi White 50 μM

**Fig 1. Examples of different modes of yeast (*S. cerevisiae*) colony growth.** (a) A small filamentous colony grown in a low assimilable nitrogen (nutrient) environment. (b) A large filamentous colony in a high nitrogen environment. (c) A different strain of yeast under low nutrient conditions.

elongating and budding end-to-end, resulting in long, thin branches of cells emanating from the centre of the colony, similar to Fig 1a. Recent work by [15] confirmed that pseudohyphal growth, not DLG, was the mechanism for non-uniform patterning in filamentous yeast colonies. However, a detailed quantitative link between the cellular mechanisms of pseudohyphal growth and macroscale patterns has not yet been established.

We use agent-based mathematical modelling to simulate filamentous colony growth. Agent-based models (ABMs) [10, 11, 16–23] track movement, proliferation, and death of individual cells in the colony. Lattice-based ABMs, such as cellular automata, restrict the positions of cells to discrete sites. Lattice-based models have been used to investigate budding patterns [24] and the role of nutrient level on colony roughness [19]. [18] developed a lattice-based model that incorporated pseudohyphal growth by introducing a directional bias to the cell growth [18]. A weakness of lattice-based models is that it is difficult to capture microscale features within the filament. Off-lattice models [25–27] address this shortcoming by allowing cells with irregular shapes to take any position on the domain. Mathematical models have shown that rod-shaped cells promote filament formation [28], and that cell morphology is a major determinant of overall colony morphology [29, 30]. Modelling has also shown that budding angle influences the irregularity in filamentous colony morphology [31]. However, the extent to which cell behaviour influences pseudohyphal colony shape in off-lattice models has yet to be fully quantified.

We extend previous studies by developing a two-dimensional off-lattice model of pseudohyphal yeast colony growth in the absence of DLG. We represent cells as ellipses, accounting for the different sizes and aspect ratios observed in regular (sated) and pseudohyphal cells. Since the timescale for nutrient diffusion is much faster than that of cell spread, we assume that nutrient concentration will be spatially uniform, such that proliferation probability is independent of space. After developing the model, we compare simulations with experiments, and infer model parameters using spatial statistics, image processing and approximate Bayesian computation [32]. Crucially, we incorporate cell size and budding angle differentiation depending on whether a cell is sated or pseudohyphal. Modelling this microscopic detail helps reveal how colony shape arises from these cell-scale behaviours. We use three spatial statistics to quantify the morphology of simulated and experimental colonies. Yeast colony morphology was primarily influenced by the colony size at the transition from sated to psuedohyphal budding, and a forking mechanism that controls formation of chains of pseudohyphal cells.

## Biological background

The biology of filamentous yeast colonies informs our mathematical model. Budding yeasts reproduce asexually by mitosis [13]. To produce new cells, an existing mother cell creates a copy of itself from a protruding bud on its outer surface. The new daughter cell replicates the genetic material of the existing cell, and eventually detaches from the mother [33]. Cells can reproduce multiple times, and each division event creates a bud scar on the mother and daughter cell surfaces. These scars are the ring-like surface structures present in Fig 2a. When not subject to external stressors, yeasts proliferate in an axial or bipolar budding pattern [34]. In axial budding, cells first bud close to the bud scar, and subsequent buds form close to previous budding sites [35]. In bipolar budding, cells reproduce at opposite ends of their major axis, and subsequent buds form at the opposite end to the initial bud [36]. We refer to a yeast cell growing in either the axial pattern or the bipolar pattern as *sated*.

Yeasts can change their growth patterns in response to stresses from the local environment [37]. These stresses include low nutrient concentration or low availability of assimilable nitrogen [38]. Under stress, yeasts do not undergo axial or bipolar budding but instead exhibit the

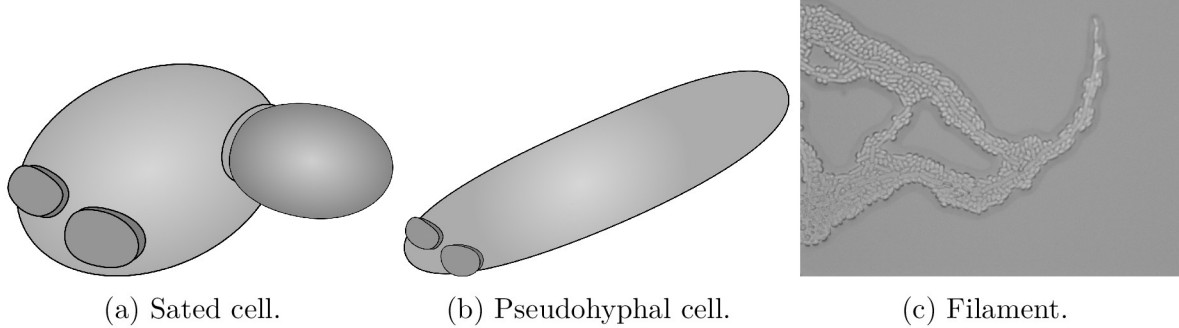

(a) Sated cell. (b) Pseudohyphal cell. (c) Filament.

**Fig 2. Cell and colony-scale behaviour in filamentous yeast colonies [13].** (a) Schematic of a sated mother cell with two bud scars and a developing daughter cell under normal conditions. (b) Schematic of an elongated pseudohyphal cell. (c) A filamentous branch at the edge of a yeast colony, consisting of both sated and pseudohyphal cells.

distal unipolar budding pattern [39]. Unipolar budding involves cells budding only at the pole directly opposite the initial bud scar. Under stress, unipolar budding is accompanied by cell elongation that increases the aspect ratio from approximately 1.5 to approximately 3.5 [13]. A typical cell elongation is shown in Fig 2b. The combination of cell elongation and unipolar budding generates longer branches emanating from the centre of the yeast colony and protruding outwards. These structures are known as pseudohyphae, as shown in Fig 2c. Pseudohyphal growth is thought to allow sessile colonies to forage for nutrients away from the densely-occupied central region [13], or to invade a host organism to more easily obtain nutrients. Pseudohyphal growth occurs in the baker's yeast *Saccharomyces cerevisiae*, the pathogen *Candida albicans* (both true hyphae and pseudohyphae), and many other yeast species.

Yeast colony behaviour is commonly studied by observing the growth of cells placed on a growth medium solidified with agar [13, 40–42], as shown in Fig 3. In the absence of stressors,

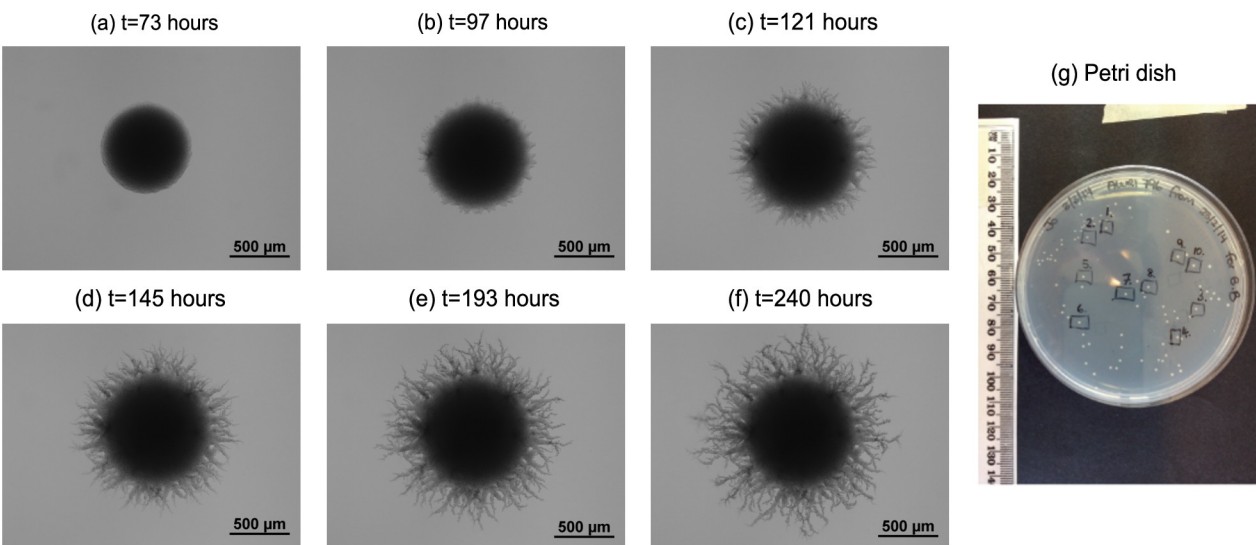

**Fig 3. Spatiotemporal evolution of a filamentous yeast colony of AWRI 796 50 μm.** (a)–(f) Time series of images from a yeast growth experiment, showing the transition from Eden-like circular growth to filamentous growth. (g) Petri dish with many concurrent filamentous colony growth experiments, indicating the small scale of filamentous colonies.

growth is approximately uniform, and yeast colonies appear circular when viewed from above. In contrast, pseudohyphal growth produces filaments that grow out from the colony, so that the growth is no longer uniform, as per Fig 3c–3f. These filaments consist of central chains of pseudohyphal cells interspersed with sated cells, as shown in the experimental image of an *S. cerevisiae* filament in Fig 2c. The length, width and taper of the filaments vary with strain and growth conditions, and the mechanisms that control these properties are not fully characterised.

## Materials and methods

We combine yeast growth experiments, mathematical modelling and simulation, image processing, and Bayesian statistical inference to investigate filamentous yeast growth. The subsections below outline our methods for each.

### Yeast growth experiments

The *Saccharomyces cerevisiae*, diploid prototrophic wine yeast strains Maurivin AWRI 796 and Enoferm Simi White were used in the experiments. Cultures were originally grown in YPD (2% glucose, 2% bactopeptone, 1% yeast extract) from commercial packets and maintained as glycerol stocks as per standard procedures. Yeast Nitrogen Base (YNB) liquid medium was prepared using dehydrated culture medium as per the manufacturer's instructions (Becton Dickson; Cat No. 233520) with the addition of glucose (2%) and ammonium sulfate (50 or 500 μM) as the sole nitrogen source, or as described previously for 'carbon base for nitrate assimilation test' [43] with the following modifications; glucose (2%), inositol (11.7 mg/L), and ammonium sulfate (50 μM or 500 μM). YNB agar plates were prepared with concentrated media stocks (2×), which were filter sterilised before mixing with an equal volume of Bacto agar (4%) (Becton Dickinson), that was previously washed twice in ultrapure water and autoclaved to sterilise. Aliquots (20 mL) were poured into standard 90 mm polystyrene Petri dishes. The yeast were cultured from glycerol stocks in 2 mL YNB (50 μM ammonium sulfate) for two days, at 28°C, with agitation. Dilutions, calculated to contain between 50 to 100 cells, were then spread onto YNB agar with 50 or 500 μM of ammonium sulfate and incubated at 28°C to yield single colonies. At least 14 yeast colonies from each condition were imaged over the timecourse of the experiments and then at the terminal point, after about 235 hours. Bright field images were viewed at 40× magnification using a Nikon Eclipse 50i microscope and imaged using a Digital Sight DS-2MBWc camera and NIS-Elements F 3.0 imaging software (Nikon).

### Mathematical model

We develop a two-dimensional off-lattice agent-based model for a filamentous yeast colony. We index cells with a subscript $i$, for $i = 1, \ldots, n$, where $n$ is the number of cells in the colony. Each cell is represented by an ellipse with centre $m_i$, major radius $a_i$, minor radius $b_i$ (with corresponding aspect ratio $d_i = a_i/b_i$), and orientation $\theta_i$. The orientation angle $\theta$ is measured anticlockwise from the $x$-axis to the major axis of the cell. The distal pole of cell $i$ is the point $m_i + v_i$, where $v_i = a_i(\cos \theta_i, \sin \theta_i)$, and the proximal pole is the point $m_i - v_i$. The size, position, and orientation of each cell in 2D are thus completely specified by five quantities: the centre $m_i = (m_x, m_y)_i$, direction vector $v_i = (v_x, v_y)_i$ and aspect ratio $d_i$. The boundary of cell $i$ is then

$$x_i(\phi) = m_i + (a_i \cos \phi \cos \theta_i - b \sin \phi \sin \theta_i, a_i \cos \phi \sin \theta_i + b_i \sin \phi \cos \theta_i), \qquad (1)$$

where $\phi \in [0, 2\pi)$ is an angle parameter such that $\phi = 0$ corresponds to the distal pole. The yeast cell model is illustrated in Fig 4a. Each cell is classified as either sated or pseudohyphal as shown in Fig 4b. Average dimensions for both cell types are known [13], and we assume they

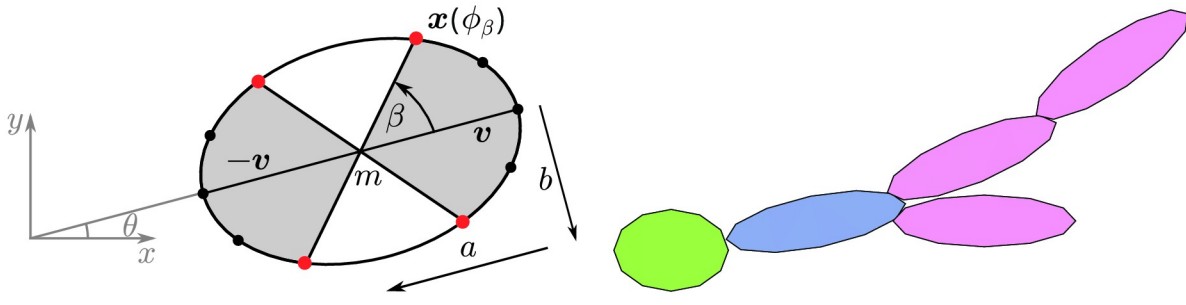

(a) Yeast cell parameterisation.                    (b) Sated and pseudohyphal cells.

**Fig 4. The yeast cell model and budding patterns.** (a) Yeast cells are represented as ellipses with centre $m$, major radius $a$, minor radius $b$ and the angle between the $x$-axis and the major axis $\theta$. The distal pole is $m + v$ and the proximal pole $m - v$. New cells are produced at bud sites (red dots) along the border of the cell, which are located between angles of $\pm\beta$ at both ends of the cell. (b) The sated (green) cell gives birth to a daughter cell (blue), which in turn yields more pseudohyphal cells (purple). Environmental stressors trigger a transition from colonies containing only sated cells to pseudohyphal growth.

do not depend on yeast strain and initial nutrient concentration. These parameter values are presented in Table 1. Importantly, with these parameters sated and pseudohyphal cells have areas that differ by 1%. Therefore, we assume that cell number can provide a good approximation for colony size, regardless of colony composition.

Cells reproduce by budding from any of four prescribed bud sites located around the boundary at each end of the cell, illustrated by red dots in Fig 4a. The bud sites at each end are equidistant and centred on the pole between the angles $\pm\beta$, relative to the cell axis. When a sated cell divides, the budding site for the daughter cell is selected at random from the four possibilities, regardless of whether the daughter is sated or pseudohyphal. In contrast, pseudohyphal cells can only bud from either of the distal poles. For a pseudohyphal mother, the budding site for the daughter cell is chosen randomly from the two potential distal sites, regardless of the daughter cell type. In terms of the cell boundary parameterisation Eq (1), a line at angle $\beta$ relative to the cell axis intersects the boundary at the point $x(\phi_\beta)$. The angle $\beta$ and parameter $\phi_\beta$ are related by

$$\tan\beta = \frac{b}{a}\tan\phi_\beta. \tag{2}$$

Therefore, for a given budding angle $\beta$, the angle parameter is

$$\phi_\beta = \beta + \frac{\arctan\left([a-b]\tan\beta\right)}{b + a\tan^2\beta}. \tag{3}$$

The angle parameter Eq (3), together with orientation $\theta_\beta$ and known $m$, $a$, and $d$ for the daughter cell, then fully characterises the boundary of the daughter cell.

We now outline the key assumptions governing colony evolution. We initiate simulations on the two-dimensional domain $(x, y) \in [-L_x/2, L_x/2] \times [-L_y/2, L_y/2]$, with a single cell with

**Table 1. The cell half width $a$, cell half length $b$ and budding angle $\beta$ for sated and pseudohyphal cells [13].**

| Parameters | Symbol | Sated | Pseudohyphal |
| --- | --- | --- | --- |
| Cell half width ($\mu m$) | $a$ | 3 | 1.9 |
| Cell half length ($\mu m$) | $b$ | 4.2 | 6.7 |
| Budding angle | $\beta$ | $7\pi/16$ | $\pi/16$ |

centre at the origin and orientation $\theta_1 = 0$. Throughout a simulation, yeast cells do not change shape or die. We neglect cell death because yeasts have lifespans of up to 10 days [44], and in low-nutrient environments can enter a stationary phase instead of dying [45]. We permit the boundary of a new cell to partially overlap an existing cell, but abort cell proliferation if the centre of the new cell would lie within the boundary of an existing cell. This partial volume exclusion enables some cell compression as cells proliferate in crowded environments. Cell overlap may also approximate cell stacking in the direction normal to the plane of the simulation, which is also observed in experiments [46]. Our volume exclusion assumption implements cellular competition without explicitly modelling the mechanical forces between cells. Neglecting mechanics allows us to focus on modelling the hypothesis that changes to cellular shape give rise to filamentous growth at the colony level. As Fig 3g shows, colonies have small size relative to the entire Petri dish. Since nutrient diffusion is very fast compared to cell proliferation in these small colonies [15], spatial variation in nutrient concentration will be negligible. Therefore, other than volume exclusion there is no spatial bias to cell proliferation. Furthermore, in our simulations we neglect time, which will not affect the eventual colony morphology. Instead, we simulate cell division events until the colony attains a prescribed number of cells, $n_{max}$. We choose this prescribed cell number $n_{max}$ such that the simulation has approximately the same area as an experimental colony, facilitating comparison between the spatial patterns in the model and experiments.

In simulations, we first select a cell to proliferate at random, and then generate a daughter cell at a budding site governed by the relevant budding pattern. Environmental stressors are known to govern the transition from sated to pseudohyphal growth. However, these stressors, such as the nutrient concentration, are difficult to measure during the experiment. Therefore, in simulations we assume that all cells are sated and produce sated offspring until the number of cells in the colony, as a proportion of the maximum, exceeds a threshold value. That is, all mother and daughter cells are sated until $n/n_{max} > n^*$, where $n^* \in (0, 1)$. This results in Eden-like circular growth, as in Fig 3a, early in the colony development. Once $n/n_{max} > n^*$, we assume sufficient environmental stress to possibly trigger pseudohyphal growth. In a simulation, if $n/n_{max} > n^*$ we implement a series of steps to propose the next proliferating cell, and the type of daughter cell produced. We outline these below, and summarise them in a decision tree, Fig 5.

The relative proliferation rates of sated and pseudohyphal cells are not precisely known. To account for this in the model, when $n/n_{max} > n^*$ we propose that the proliferating cell is sated with probability $p_a$, where $p_a$ is a parameter that we can vary. This models the idea that a transition to pseudohyphal growth is a colony-level decision, rather than one made by individual cells. This assumption acknowledges that there are complex colony-level genetic and signalling pathways that govern pseudohyphal growth, without modelling them explicitly [14]. A pseudohyphal cell is selected to proliferate with the complementary probability $1 - p_a$. The only exception to this rule is if the yeast contains no pseudohyphal cells. If so, we must select a sated cell. Given $n/n_{max} > n^*$, a sated cell will produce a pseudohyphal daughter with probability $p_{sp}$, and a sated daughter cell with probability $1 - p_{sp}$.

If the cell chosen to proliferate is pseudohyphal, we also check whether the cell to proliferate has an existing pseudohyphal daughter. If not, the daughter cell is pseudohyphal, in accordance with end-to-end unipolar budding of pseudohyphal cells. However, if the pseudohyphal cell chosen to proliferate already has an existing daughter cell, its behaviour is controlled by two parameters, $\gamma$ and $p_{ps}$. The parameter $\gamma$ limits forking from pseudohyphal cells, which is the scenario where multiple pseudohyphal branches emanate from the same pseudohyphal mother. Once a pseudohyphal cell has been chosen to proliferate, with probability $\gamma$, we abort proliferation, and instead select a new pseudohyphal mother cell with no pseudohyphal

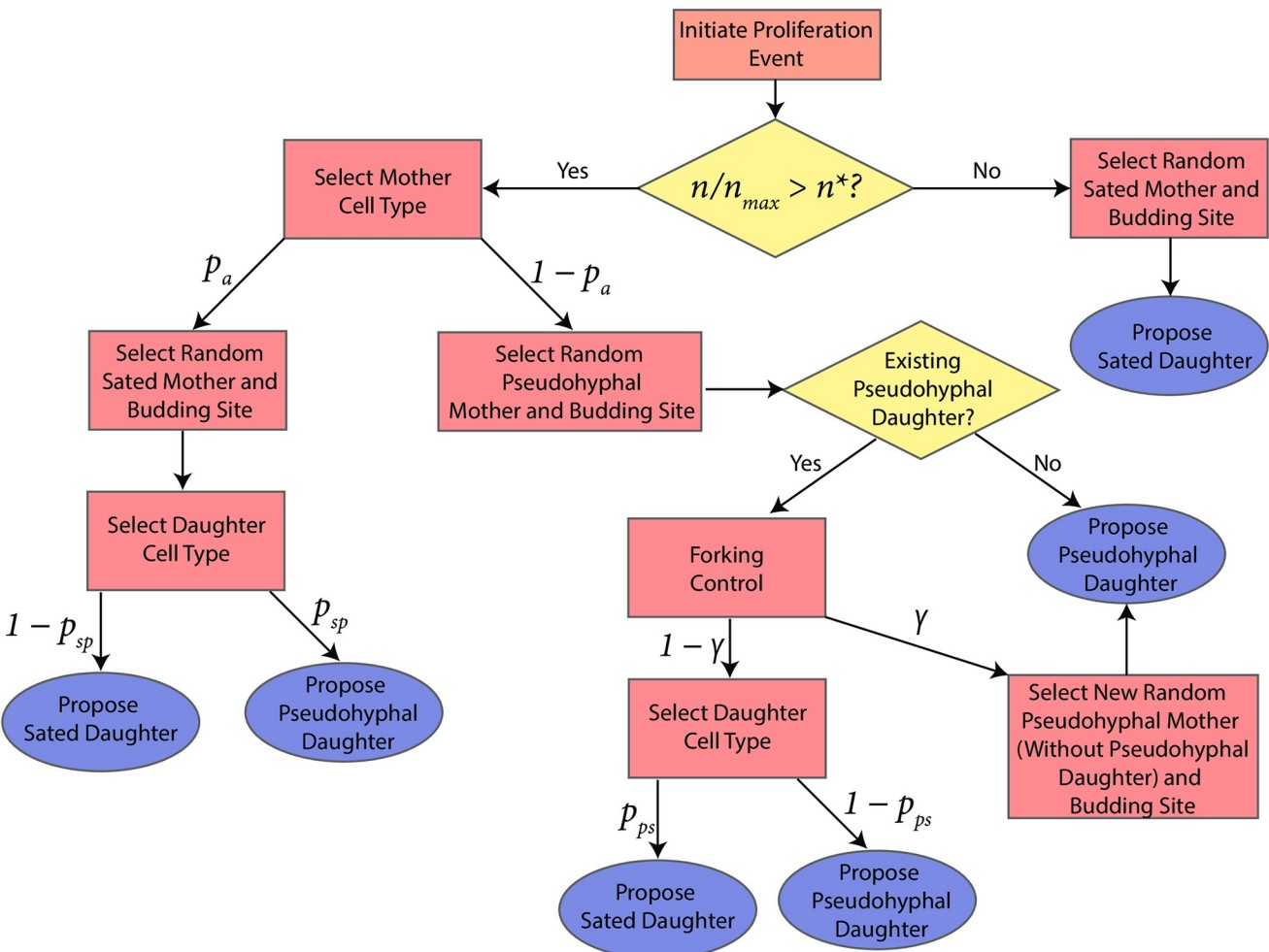

**Fig 5. Flow chart outlining the process for deciding which type of new cell to propose when a cell proliferates.** When selecting a cell type, the probability of each event is indicated using the symbols that appear above the arrows.

daughter to proliferate. Comparatively large $\gamma$ models the scenario where the colony preferentially chooses to generate a new filamentous branch, increasing its ability to forage radially. Alternatively, with probability $1 - \gamma$, the chosen pseudohyphal cell with an existing pseudohyphal daughter is allowed to proliferate. In this scenario, the daughter cell can be either sated, with probability $p_{ps}$, or pseudohyphal, with probability $1 - p_{ps}$. Enabling both daughter cell types is in accordance with experimental observations of filaments (see Fig 2c), where sated and pseudohyphal cells coexist. Finally, after proposing any proliferation event, if no budding site is available or the event contradicts our volume exclusion condition, the event is aborted, and the procedure begins again from the start. Producing an entire colony involves repeatedly deciding on cell proliferation events until the colony attains a desired target area. As summarised in Table 2, our off-lattice ABM contains five parameters: $n^*$, $p_a$, $p_{sp}$, $p_{ps}$, and $\gamma$. Since parameter values are difficult to determine in experiments, we infer them by comparing simulations with experimental photographs. We present image processing and parameter inference methods in the sections below.

**Table 2. Parameters that we vary in the off-lattice model.** All parameters in this table take values in the interval [0, 1].

| Symbol | Description |
|--------|-------------|
| $n^*$ | Proportion of total colony growth above which we permit pseudohyphal growth. |
| $p_a$ | Probability of selecting a sated mother cell if $n/n_{\max} > n^*$. |
| $p_{sp}$ | Probability of a sated mother cell producing a pseudohyphal daughter cell. |
| $p_{ps}$ | Probability of a pseudohyphal mother cell producing a sated daughter cell. |
| $\gamma$ | Forking control parameter for pseudohyphal daughter cells. |

## Image processing and summary statistics

Image processing and quantification enable us to compare experimental and simulation results. The first step is to convert grayscale photographs and simulation plots to binary images. In all images, dark colours indicates regions occupied by the colony, whereas light colour indicates unoccupied regions. For an image with $X \times Y$ pixels labelled as the ordered pairs $(x_i, y_j)$, for $i = 1, \ldots X$ and $j = 1, \ldots Y$, a binary image is a matrix

$$M_{i,j} = \begin{cases} 1 & \text{if} \quad (x_i, y_j) \quad \text{occupied} \\ 0 & \text{if} \quad (x_i, y_j) \quad \text{unoccupied.} \end{cases} \tag{4}$$

We obtain the binary images using the Tool for Analysis of the Morphology of Microbial Colonies (TAMMiCol) [47], which uses automatic thresholding and interpolation to obtain binary images. We save simulation images at the same pixel resolution as experimental photographs, such that binary images Eq (4) enable direct comparison between simulations and experiments.

Summary statistics for the spatial pattern enable systematic comparison between simulated and experimental images. In this work, we compute three summary statistics, which we use to infer model parameters based on experimental observations. These summary statistics are the sub-branch count ($I_B$), filamentous area ratio ($I_F$), and radius ratio ($I_R$). These summary statistics quantify the extent of filamentous growth in a colony, and the spatial distribution of filaments. The radius ratio is defined as $I_R = r_{csr}/r_{\max}$, where $r_{csr}$ and $r_{\max}$ are the complete spatial randomness (CSR) radius and maximum radius of the colony [42], respectively. To compute these radii, we first use MATLAB to locate the colony centroid. The maximum radius, $r_{\max}$, is the maximum distance from the centroid to an occupied pixel. The CSR radius, $r_{csr}$, is the distance between the centroid and the radial position where the density of occupied pixels is equal to $\rho_{csr} = N/\pi r_{\max}^2$, where $N = \Sigma_{i,j} M_{i,j}$ is the number of occupied pixels in the colony. Fig 6b illustrates these radii for the colony in Fig 3f. The radius ratio, $I_R$, then indicates the proportion of the colony occupied by the central circular region of Eden-like growth.

The filamentous area ratio, $I_F$, is the ratio of the number of occupied pixels in the annular region between $r_{csr}$ and $r_{\max}$, to the total number of occupied pixels in the colony. Like the radius ratio, $I_F$ quantifies the extent of filamentation but also accounts for the density of filaments in addition to their maximum length. The sub-branch count, $I_B$, characterises the shape of filaments in the angular direction. MATLAB's `bwmorph()` function provides a count of the number of branches in a skeletonised image of the filamentous region. An example skeletonised image is shown in Fig 6c. To obtain the sub-branch count, we count the number of branches of a skeletonised experimental image. Due to the computational cost of skeletonisation, we only counted a quarter of the colony's branches. We used the top left corner, counting

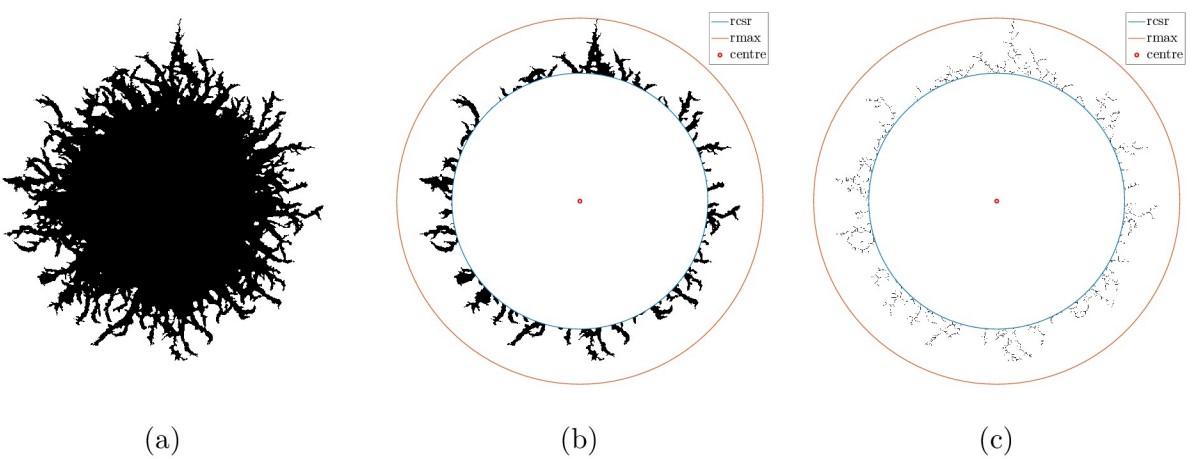

(a)                              (b)                              (c)

**Fig 6.** (a) Processed image of the experimental yeast colony in Fig 3f. (b) A binary image of the colony with the occupied central region removed from the image. The red marker is the colony centroid, such that the blue and orange curves indicate $r_{csr}$ and $r_{max}$ used to compute the radius ratio. (c) Skeletonised image of (b) obtained using MATLAB's `bwmorph()` command. This image is used to compute the sub-branch count summary statistic.

branches in the region $(r, \theta)$ where $r \in [r_{csr}, r_{max}]$, and $\theta \in [\pi/2, \pi]$, where $r = 0$ represents the colony centroid.

## Parameter inference

We perform Bayesian statistical analysis to infer posterior distributions for the model parameters, $\theta = (n^*, p_a, p_{sp}, p_{ps}, \gamma)$, associated with different experimental images [48]. However, due to the complexity of the model, which is specified implicitly as a simulation procedure, computing the likelihood of the data, given the parameters, is intractable. To overcome this issue, we use approximate Bayesian computation (ABC) [32]. This class of algorithms samples from an approximation to the true posterior by using simulations of the model and comparing these with the observed images (the data). The benefit of the ABC approach is that, instead of capturing only point estimates of the parameters, we infer their full joint posterior distribution and thus capture the variability and covariance structure as well.

As our model has five parameters, we choose to implement a version of ABC that uses Metropolis-Hastings (MH) steps [49, 50], rather than rejection sampling. This works by simulating a Markov chain whose stationary distribution is approximately the same as the target posterior distribution. At each iteration of the chain, a new set of parameters, $\theta^*$, are proposed based on the current set, $\theta_i$, and these are accepted or rejected based on how 'close' the simulated images are to the experimental results. More specifically, iterations are accepted with probability,

$$\alpha = \min\left\{1, \frac{\pi(\theta^*)q(\theta_i|\theta^*)}{\pi(\theta_i)q(\theta^*|\theta_i)}\mathbf{1}\{\rho(S(x), \bar{S}) \leq \epsilon\}\right\}, \tag{5}$$

where $\pi(.)$ is the prior, $q(.)$ is the proposal density, $\mathbf{1}(.)$ is the indicator function, $\rho(.)$ is a distance metric, $S(.)$ is a function returning a vector of summary statistics, $x$ is an image of the colony from a simulation, and $\bar{S}$ denotes mean summary statistics from the experimental data. The dependence on the indicator function in Eq (5) means that for a given simulation, if $\rho(.) > \epsilon$, where $\epsilon$ is a chosen error tolerance, then the indicator is zero and hence so is $\alpha$ and therefore the proposal is rejected. We chose a multivariate Gaussian for the proposal

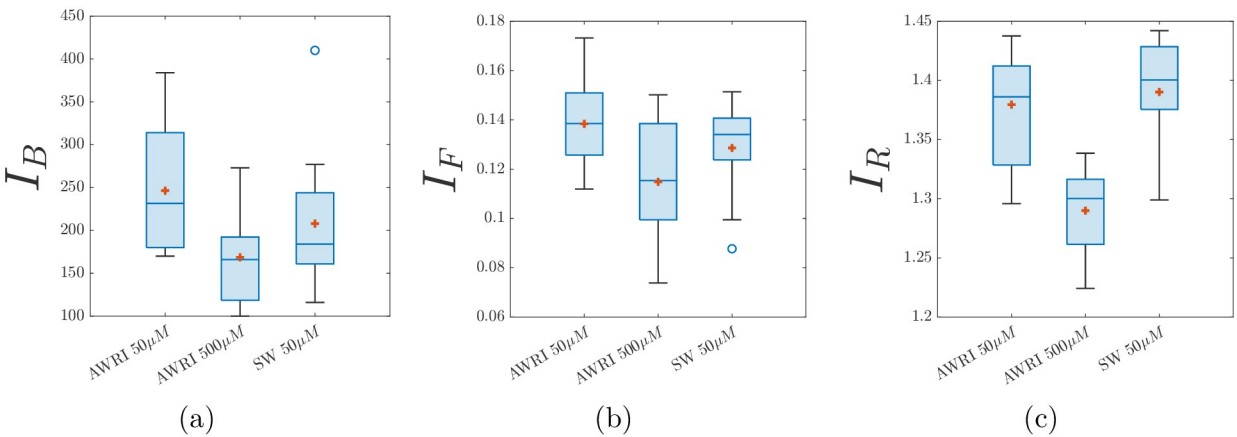

**Fig 7. Box plots of metrics to distinguish between experiments with varying nutrient (ammonium sulfate) concentrations and strains.** The red marker represents the sample mean. (a) Sub-branch count, $I_B$, (b) Filament area ratio, $I_F$, and (c) Radius ratio, $I_R$. SW: Simi White.

distribution, which is symmetric; hence, the proposal densities, $q(.)$, cancel out in the acceptance probability.

We quantify the closeness between a simulation and an experiment using a distance metric that incorporates the three summary statistics described in the Image Processing and Summary Statistics section. However, typical values of $I_B$ are much larger than $I_F$ and $I_R$, as Fig 7 shows. Hence, we define $S(x)$ and $\bar{S}$ in terms of normalised summary statistics to weight the contributions of sub-branch count, filamentous area ratio, and radius ratio equally. The normalisation for each statistic is carried out relative to vectors containing summary statistics from all images of the same experimental conditions, denoted $\boldsymbol{I_B}$, $\boldsymbol{I_F}$, and $\boldsymbol{I_R}$. These vectors are of length $N$, the number of replicates of a given experiment. Then, given a summary statistic vector $(I_B(x), I_F(x), I_R(x))$ for a single simulation image $x$, we can normalise as

$$S(x) = \left[ \frac{I_B(x) - \min(\boldsymbol{I_B})}{\max(\boldsymbol{I_B}) - \min(\boldsymbol{I_B})}, \frac{I_F(x) - \min(\boldsymbol{I_F})}{\max(\boldsymbol{I_F}) - \min(\boldsymbol{I_F})}, \frac{I_R(x) - \min(\boldsymbol{I_R})}{\max(\boldsymbol{I_R}) - \min(\boldsymbol{I_R})} \right], \quad (6)$$

where $S(x)$ is the normalised summary statistic vector, such that each element of $S(x)$ takes values near the unit interval. We also define

$$\bar{S} = (\tilde{I}_B, \tilde{I}_F, \tilde{I}_R) = \left[ \frac{\bar{\boldsymbol{I}}_B - \min(\boldsymbol{I_B})}{\max(\boldsymbol{I_B}) - \min(\boldsymbol{I_B})}, \frac{\bar{\boldsymbol{I}}_F - \min(\boldsymbol{I_F})}{\max(\boldsymbol{I_F}) - \min(\boldsymbol{I_F})}, \frac{\bar{\boldsymbol{I}}_R - \min(\boldsymbol{I_R})}{\max(\boldsymbol{I_R}) - \min(\boldsymbol{I_R})} \right], \quad (7)$$

where a bar denotes the mean of an experimental data vector. We then use the normalised Euclidean distance metric, $\rho(.)$, given by [50],

$$\rho(S(x), \bar{S}) = ||S(x) - \bar{S}|| = \frac{1}{3} \left[ \sum_{i=1}^{3} \left( \frac{S(x)_i - \bar{S}_i}{1 + \bar{S}_i} \right)^2 \right]^{\frac{1}{2}}, \quad (8)$$

We chose to compare simulations with the mean of the summary statistic vector, $\bar{S}$, for each given experimental condition rather than for each individual replicate. Despite this simplification, this was found to produce simulated colonies that closely resemble the experimentally observed morphology. Inferring parameters for single experimental images is also possible, but would not necessarily yield accurate posterior distributions due to wide variation across experiments.

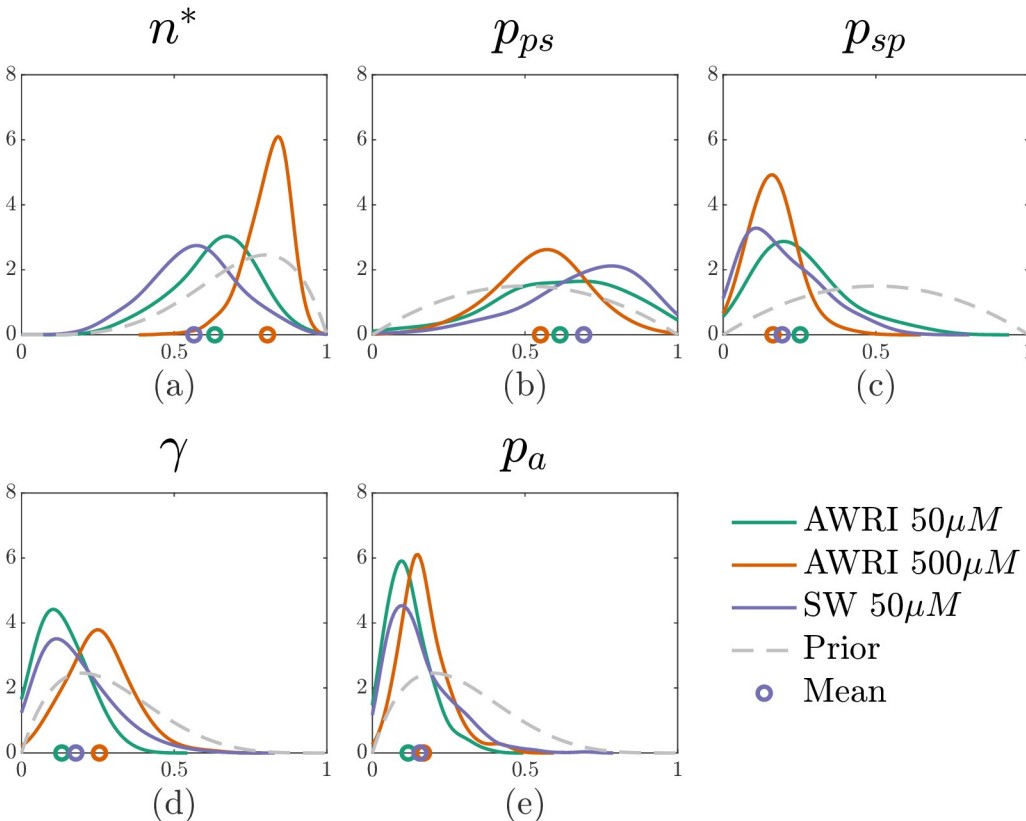

**Fig 8. Posterior densities for average colony summary statistics and mean colony areas of AWRI 796 50 µM, AWRI 796 500 µM, and Simi White (SW) 50 µM.** The marker below the distribution shows the mean. The priors are as follows (a) $n^* \sim$ Beta(5, 2), (b) $p_{ps} \sim$ Beta(2, 2), (c) $p_{sp} \sim$ Beta(2, 2), (d) $\gamma \sim$ Beta(2, 5), and (e) $p_a \sim$ Beta(2, 5).

We chose Beta distribution priors (see Fig 8), and the proposal covariance matrix is calculated from doing pilot runs with a higher value of $\epsilon$ to encourage faster mixing. We chose weakly informative priors for $n^*$, $\gamma$, and $p_a$, and uninformative priors for $p_{ps}$ and $p_{sp}$. We use weakly informative priors when physical intuition guides our expectation for the values these parameters may take, whereas less is known about $p_{ps}$ and $p_{sp}$. We expect $n^*$ to be relatively large, because experimental images show pseudohyphal growth emerging after a long initial period of circular (Eden-like sated) growth. The parameter $p_a$ is likely to be small, because after the transition to pseudohyphal growth is initiated, we expect most of the growth to be pseudohyphal rather than sated. Finally, we expect the forking control mechanism controlled by $\gamma$ to be invoked infrequently, because experimental images suggest that pseudohyphal branches do fork regularly. A second reason for using weakly informative priors is that they help the performance of the ABC parameter inference algorithm. The chain can become stuck (proposed moves are rarely accepted) if it enters regions of the parameter space where the simulated and experimental data are very different, and weakly informative priors can help prevent this. However, we did explore using uniform priors and show results in Fig C in S1 Text. There was little difference in the mean of the posterior distribution, suggesting our weakly informative priors were appropriate.

With the covariance matrix fixed, we then decreased the error tolerance until the acceptance rate of the chain was between 5–10% [51]. Trace plots were employed to assess the convergence and mixing of the chains (see Fig A in S1 Text). As the simulation of the off-lattice

**Table 3. Mean and standard deviation of the normalised summary statistics for experimental images, where the normalisation is based on Eq (7).**

|  | $\tilde{I}_B$ | $\tilde{I}_F$ | $\tilde{I}_R$ |
|---|---|---|---|
| AWRI 50 μM | 0.49 ± 0.20 | 0.62 ± 0.17 | 0.69 ± 0.20 |
| AWRI 500 μM | 0.26 ± 0.16 | 0.41 ± 0.25 | 0.30 ± 0.16 |
| Simi White 50 μM | 0.38 ± 0.22 | 0.53 ± 0.16 | 0.74 ± 0.16 |

model is computationally expensive, we ran multiple chains independently and concatenated the samples to reduce total computation time. For each parameter, we used an initial value a small perturbation away from the mean of the prior distribution. We ran each chain for a sufficient time until an effective sample size (ESS) of at least 200 was reached. The ESS values and the tuning parameter, $\epsilon$, for each experimental setup are given in Table A in S1 Text.

## Results and discussion

Below we present the inference results and compare simulated colonies using parameters drawn from the posteriors with the experimental images. This allows us to gain insight into the driving factors in the cell biology of the shape formation of the yeast *S. cerevisiae*.

### Experimental results

Overall, the summary statistics can distinguish between experiments based on initial nutrient concentration and yeast strain. For example, Fig 7 shows that increasing initial nutrient concentration decreases all three metrics, indicating less filamentous growth. Furthermore, we observe a clear difference in the mean of the summary statistics between experimental conditions, which are used in the inference. These results suggest that the summary statistics are suitable for quantifying some of the differences in colony morphology. A tabulated version of the means and standard deviations of the normalised summary statistics of the experimental colonies is given in Table 3.

### Simulation and inference results

Kernel density estimates of the marginal posteriors for the five model parameters for all three experiments are shown in Fig 8. A key distinction between the three experiments is the value of $n^*$, which is the proportion of total colony growth before pseudohyphal growth is permitted (see Fig 8a). As expected, colonies at a lower initial nutrient level (50 μM) have a smaller proportion of sated cells produced before pseudohyphal growth is permitted compared to colonies under higher initial nutrient levels (500 μM). According to our parameter inference, the number of cells in the colony before pseudohyphal growth commences, $n^* n_{max}$, is larger by a factor of approximately 2.6 in the 500 μM initial nutrient AWRI colony compared to the 50 μM initial nutrient AWRI colony. Therefore, although our results reflect that low nutrient concentration is an environmental stressor that triggers psuedohyphal growth, $n^* n_{max}$ is not proportional to the initial nutrient concentration. A possible explanation is that sustaining a larger colony requires more nutrients, such that larger colonies have a higher critical nutrient threshold than smaller colonies. A second key distinction reflected in our model is $\gamma$, the probability that controls the reproduction of a second pseudohyphal daughter cell as displayed in Fig 8d. This parameter can be interpreted as controlling the forking mechanism within colony branches. A smaller $\gamma$ means a higher probability of cells having two pseudohyphal daughter cells. We

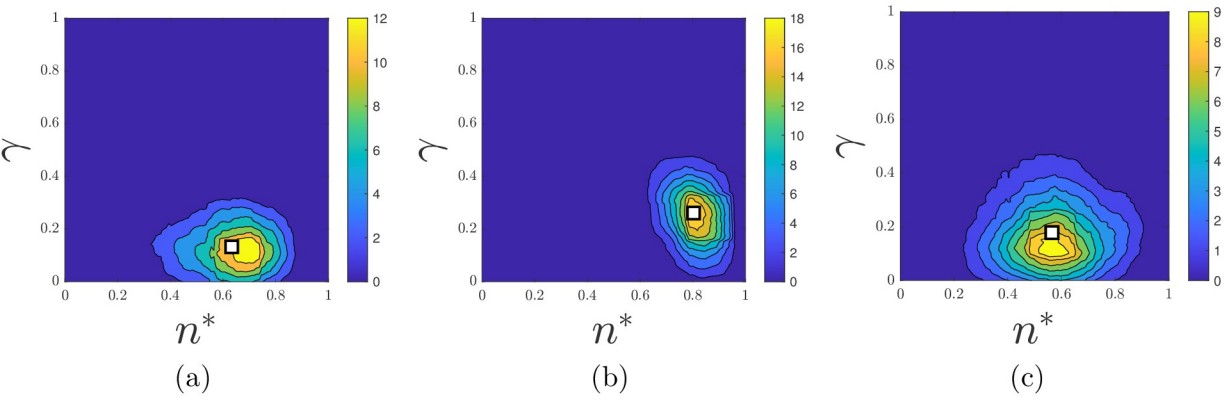

**Fig 9. Posterior bivariate densities of $\gamma$ vs $n^*$ for (a) AWRI 50 µM, (b) AWRI 500 µM and (c) Simi White 50 µM.**

observe that colonies of lower initial nutrient concentration exhibit more forking than the higher initial nutrient colonies (see Fig 10).

To highlight the impact of these two parameter values, we also plotted bivariate kernel density estimators of these two parameter values as shown in Fig 9. The AWRI colony with 500 µM initial nutrient corresponds to larger values of $n^*$ and $\gamma$ than the colonies with 50 µM initial nutrient. Larger $\gamma$ lessens forking, in accordance with Fig 10. We emphasise that although forking is crucial to reproducing the observed experimental morphology in simulations, we do not explicitly model the biological drivers of differentiated forking behaviour. Our work suggests that understanding the forking mechanism would be a useful target for future experimental work, with the goal of controlling colony morphology. Another interesting observation is that the initial nutrient concentration has a greater impact on $n^*$ and $\gamma$ than yeast strain. Therefore, initial nutrient levels are an essential determinant of colony morphology, whereas yeast strain makes a more subtle difference.

Focusing on the probabilities $p_{ps}$ and $p_{sp}$, at higher initial nutrient concentrations colonies have a lower probability of producing pseudohyphal cells compared to the colonies with lower initial nutrient concentrations, as presented in Fig 8b and 8c. Filament length is likely to be caused by chains of pseudohyphal cells, and colonies in higher nutrient environments have shorter filaments. On the other hand, the posterior for the parameter controlling the probability of transition from pseudohyphal to sated ($p_{ps}$) was uninformative (the posterior is similar to the prior) as shown in Fig 8b. One possible explanation is that $p_{ps}$ has little impact on growth patterns because a sated or pseudohyphal cell might occupy a similar area within a filament. Similarly, the parameter $p_a$, which is the probability of biasing pseudohyphal daughter cell reproduction, could not be distinguished among the three experiments either (see Fig 8e). One explanation for this result could be that pseudohyphal cell reproduction is similar across the three different types of experimental conditions.

We now present visual simulation results using the mean of the marginal posteriors, which are indicated by the round markers in Fig 8 and the mean colony area of the experiment. Results are shown in Fig 10. The simulations are quantitatively similar in size compared to the experiments (pixel area), as per the model design. In addition, the simulated images closely resemble the experimental colonies. However, since our model is stochastic, every simulation of the same parameter values will generate different colonies. Therefore, it is impossible to capture the exact morphology to match the experiments. Moreover, since we infer parameters on the mean of several similar experimental replicates, the parameter values used in Fig 10 were

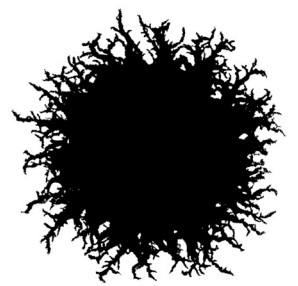

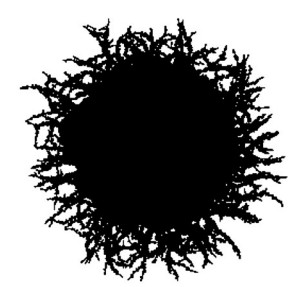

(a) AWRI 50 μM S1 experiment.  (b) Simulation (AWRI 50 μM)

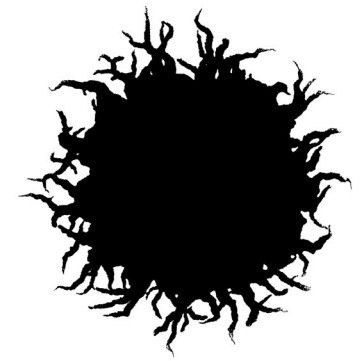

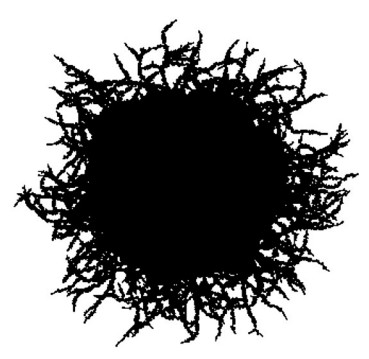

(c) Simi White 50 μM S7 experiment.  (d) Simulation (Simi White 50 μM)

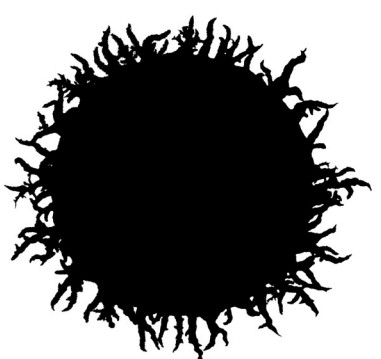

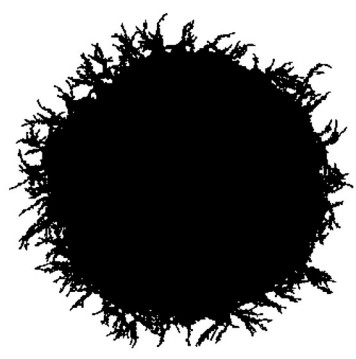

(e) AWRI 500 μM S8 experiment.  (f) Simulation (AWRI 500 μM)

**Fig 10. Comparison between experiments and simulations with parameters inferred using the mean summary statistics from all replicates of given experimental conditions.** (a) Example experiment, AWRI strain with 50 μM initial nutrient (ammonium sulfate). (b) Simulation with $\theta = (n^*, p_a, p_{sp}, p_{ps}, \gamma) = (0.67, 0.14, 0.25, 0.58, 0.12)$. (c) Example experiment, Simi White strain with 50 μM initial nutrient. (d) Simulation with $\theta = (n^*, p_a, p_{sp}, p_{ps}, \gamma) = (0.62, 0.15, 0.23, 0.71, 0.16)$ (e) Example experiment, AWRI strain with 500 μM initial nutrient. (f) Simulation with $\theta = (n^*, p_a, p_{sp}, p_{ps}, \gamma) = (0.81, 0.16, 0.19, 0.49, 0.25)$.

not directly inferred from the sample experimental replicate shown. Results for parameter inference applied to a selection of individual colonies are in Fig B in S1 Text. As expected, inferring on an individual colony also yields morphologies similar to experiments.

　　Some limitations of our study are important to consider when evaluating the results. An advantage of our approach is that we could characterise the filamentous morphology using

only three summary statistics. However, inferring five parameters from these three summary statistics might have contributed to the inconclusive results for parameters $p_{ps}$ and $p_a$. Since our model neglected time, we only compared simulations and experiments using the final photograph of the colony morphology, rather than data from multiple time points. However, observing the time-dependent colony growth in experiments and incorporating this information into the parameter inference could increase the resolution of the results. One reason not to add time dependence is that, with current techniques, the inference would be too computationally expensive, limiting the length of the chains and, hence, the ESS and the scope of our results. In future work, more advanced Bayesian computation methods could be applied, such as sequential ABC [52]. This might speed up the inference, and help to achieve higher ESS values and increase the fidelity of parameter inference. However, even with these limitations we obtain strong agreement between experimental and simulated morphologies. This reinforces the importance of initial nutrient level and cellular behaviour in driving filamentous colony growth.

## Conclusion

In this paper, we developed an off-lattice model to understand the cellular mechanisms that contribute to colony morphology of the yeast *Saccharomyces cerevisiae*. We analysed three different experiments: two with the same initial nutrient concentration but varying strains, and two with the same strain but varying initial nutrient concentration. To couple our model with experimental results, we employed an ABC method to calibrate our model to match closely with the experimental images. We do not capture all possible sources of experimental variation, such as different initial conditions, cellular mechanics, and explicit modelling of cell crowding. However, our work isolates and quantifies the effects of cell shape changes on colony morphology. The key factors contributing to the different morphology were the colony size at the sated-to-pseudohyphal transition, and a forking mechanism that controls the reproduction of a second pseudohyphal cell. Although simple, a benefit of our model is its potential adaptability to other morphologies. In this study, applied specifically to filamentous yeast colonies, the five parameters that we allowed to vary were the most important for filamentous colonies. However, in different contexts other features such as cell size and proliferation angle could also be made variable. In future work, we will adapt the model to explore other unique morphologies in colonies of yeast and other microorganisms.

## Supporting information

**S1 Text. This file includes a summary table of tuning parameters for the ABC-MCMC algorithm, ABC-MCMC trace plots diagnoses, individual simulated colonies and an investigation into uninformative priors. Table A. Effective sample size and acceptance threshold for ABC-MCMC results.** ESS and $\epsilon$ acceptance threshold for ABC-MCMC algorithm. Each iteration of the chain is simulated up to the average pixel colony area of the respective colonies, allowing up to a 5% difference. In addition, the computation time for a single colony and their average cell number are given below. Computations are performed using an Intel Xeon CPU E5–2699 v3 (2.30GHz). **Fig A. ABC-MCMC trace plots.** Trace plots of ABC-MCMC algorithm for (a) AWRI 50 μM (ESS of 205.151), (b) AWRI 500 μM (ESS of 222.225) and (c) Simi White 50 μM (ESS of 251.980). Each chain is running until 20,000 iterations. **Fig B. Individual colony simulations.** Comparison between experiments and simulations with parameters inferred using individual colonies. (a–b) AWRI strain with 50 μM nutrient. (c–d) AWRI strain with 500 μM nutrient. (e) Simulation with $\theta = (n^*, p_a, p_{sp}, p_{ps}, \gamma) = (0.56, 0.10, 0.31, 0.65, 0.16)$ inferred S12 AWRI 50 μM experiments. (f) Simulation with $\theta =$

$(n^*, p_a, p_{sp}, p_{ps}, \gamma) = (0.66, 0.10, 0.23, 0.54, 0.15)$ inferred S7 AWRI 50 μM experiments. (g) Simulation with $\theta = (n^*, p_a, p_{sp}, p_{ps}, \gamma) = (0.89, 0.09, 0.28, 0.67, 0.14)$ inferred S11 AWRI 500 μM experiments. (h) Simulation with $\theta = (n^*, p_a, p_{sp}, p_{ps}, \gamma) = (0.78, 0.08, 0.20, 0.70, 0.13)$ inferred S8 AWRI 500 μM experiments. **Fig C. ABC-MCMC trace plot with uninformative priors.** Trace plot of ABC-MCMC algorithm for AWRI 796 50 μM using uniform priors. (PDF)

## Acknowledgments

The authors acknowledge Trent Mattner for helpful discussions and assistance with the supercomputer and John Maclean for initial discussion on ABC. This work was supported with supercomputing resources provided by the maths1 High Performance Computing service at the University of Adelaide.

## Author Contributions

**Conceptualization:** Kai Li, J. Edward F. Green, Hayden Tronnolone, Jennifer M. Gardner, Joanna F. Sundstrom, Vladimir Jiranek, Benjamin J. Binder.

**Formal analysis:** Kai Li.

**Funding acquisition:** J. Edward F. Green, Alexander K. Y. Tam, Vladimir Jiranek, Benjamin J. Binder.

**Investigation:** Jennifer M. Gardner, Joanna F. Sundstrom.

**Methodology:** Kai Li, J. Edward F. Green, Hayden Tronnolone, Andrew J. Black, Jennifer M. Gardner, Joanna F. Sundstrom, Vladimir Jiranek, Benjamin J. Binder.

**Project administration:** Benjamin J. Binder.

**Software:** Kai Li, Hayden Tronnolone.

**Supervision:** J. Edward F. Green, Alexander K. Y. Tam, Andrew J. Black, Vladimir Jiranek, Benjamin J. Binder.

**Visualization:** Kai Li.

**Writing – original draft:** Kai Li, Hayden Tronnolone, Alexander K. Y. Tam.

**Writing – review & editing:** J. Edward F. Green, Andrew J. Black, Jennifer M. Gardner, Vladimir Jiranek, Benjamin J. Binder.

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
