## [Decision Letter · Decision Letter 0]

7 Sep 2024

Dear Professor Jiranek,

Thank you very much for submitting your manuscript "An off-lattice discrete model to characterise filamentous yeast colony morphology" for consideration at PLOS Computational Biology. As with all papers reviewed by the journal, your manuscript was reviewed by members of the editorial board and by several independent reviewers. The reviewers appreciated the attention to an important topic. Based on the reviews, we are likely to accept this manuscript for publication, providing that you modify the manuscript according to the review recommendations.

Please ensure you address the main points raised by the referees, especially the major concerns of reviewer 1 on the lack of modelling nutrients.

Sincerely,

Attila Csikász-Nagy

Academic Editor

PLOS Computational Biology

Daniel Beard

Section Editor

PLOS Computational Biology

Reviewer's Responses to Questions

**Comments to the Authors:**

Reviewer #1: Please see attached document.

Reviewer #2: The paper is interesting the described methods are elegant. It communicates the motivation of the work, the applied methods and the results clearly. The text is also well written.

Questions, comments:

- What were the reasons behind using the two mentioned yeast strains?

- Figure 1.3: What were the experimental conditions (strain, nutrient level) of the colonies shown at the image?

- Figure 2.1 shows 5 budding sites at the end of the cell, while line 122 mentions 3 sites. Which value was used in the model?

- Can filaments cross each other in the model? Would it possible or useful to extend the model into 3D?

- What were the typical cell numbers in the simulations?

- What kinds of computational resources were used for the presented work? How much time a typical does a typical simulation required?

- Is there any cell death in the model? How could it influence the results?

- Line 222 states: “we only counted a quarter of the colony’s branches”. Could you elaborate on this (e.g.: which quarter was counted)?

- How were the initial values of the parameters defined at the parameter inference?

- Figure 3.1: Were the differences between the showed metrics statistically significant?

- Could be any difference in the cell sizes, shapes or cell growths (division intensities) between the studied strains? - - Could these influence the differences between their colonies (on top of the studied parameters)?

- Link for the supporting materials is missing from the “Supporting Material” section. (It appears at the end of the manuscript.)

- Most of the references are older. Are there more state of the art publications on the topic?

Minor suggestions:

- Line 3: Yeasts (typo)

- Line 144: Figure1.3g

- IB is both called branch count and sub-branch index

- Line 284: means, deviations, statistics

- Figure 3.1: concentrations, strains

- Figure 3.2: statistics, areas

- Line 328: is in the

- Line 376: Credit (capital letters)

**Have the authors made all data and (if applicable) computational code underlying the findings in their manuscript fully available?**

Reviewer #1: None

Reviewer #2: Yes

PLOS authors have the option to publish the peer review history of their article (what does this mean?). If published, this will include your full peer review and any attached files.

Reviewer #1: No

Reviewer #2: No

Figure Files:

Data Requirements:

Reproducibility:

References:

---

## [Decision Letter · Decision Letter 1]

13 Oct 2024

Dear Professor Jiranek,

Thank you very much for submitting your manuscript "An off-lattice discrete model to characterise filamentous yeast colony morphology" for consideration at PLOS Computational Biology. As with all papers reviewed by the journal, your manuscript was reviewed by members of the editorial board and by several independent reviewers. The reviewers appreciated the attention to an important topic. Based on the reviews, we are likely to accept this manuscript for publication, providing that you modify the manuscript according to the review recommendations.

Referee 1 still have concerns mainly about the way nutrient levels are handled in the model. Please address these concerns in a revised version!

Sincerely,

Attila Csikász-Nagy

Academic Editor

PLOS Computational Biology

Daniel Beard

Section Editor

PLOS Computational Biology

Reviewer's Responses to Questions

**Comments to the Authors:**

Reviewer #1: I thank the authors for responding to each of my original comments. While I do find many of the responses reasonable, I feel that none of my original concerns were actually properly addressed (but rather, rebutted). The comments, in particular, that I feel need to be addressed are as follows.

1. The authors claim very early on that "initial nutrient concentration" among other features, was a key determining feature that determines the shape and structure of the colony, however they do not explicitly model nutrients. They do have experimental information that could give insight into this (two experiments with the same strain, with a 10-fold difference in nutrient concentration), with which they could do a statistical test (for instance) to determine if key parameters are different. How do the authors reconcile the 10-fold difference in nutrient concentration with similar values of n^* (as per the posterior distributions in Fig. 8)? Presumably, the authors could, for instance, use a fit to AWRI 50uM data to test their hypothesis that nutrient concentration is proportional to cell-count to predict AWRI 500uM data.

In short, I am of the view that the authors cannot state any conclusions relating to nutrient concentration without either: (1) performing some kind of statistical demonstratation to discuss how n^* is different between 50uM and 500uM measurements; (2) performing some kind of validation for the nutrient-cell count assumption; or (3) explicitly modelling the nutrient concentration.

2. I accept that ABC has known convergence issues, however the parameter space is relatively small (only five parameters). For example, ABC MCMC or even ABC SMC can give good results even for relatively uninformative priors. Investigating the effect of prior choice on at least one of the conditions would significantly strengthen the paper.

3. (relatively minor) In response to original minor comment 3, the authors state that they initialise the colonies from a single cell. Is this actually how the experiments are initiated? This could significantly impact the effect of the nutrient-cell count model, since a colony that is initiated with a set nutrient count at a size that is not zero would have additional biomass not accounted for by nutrient consumption.

Reviewer #2: Thank you for your answers and your clarifications in the manuscript. I accept them.

**Have the authors made all data and (if applicable) computational code underlying the findings in their manuscript fully available?**

Reviewer #1: Yes

Reviewer #2: Yes

PLOS authors have the option to publish the peer review history of their article (what does this mean?). If published, this will include your full peer review and any attached files.

Reviewer #1: No

Reviewer #2: No

Figure Files:

Data Requirements:

Reproducibility:

References:

---

## [Editor Report · Decision Letter 2]

3 Nov 2024

Dear Professor Jiranek,

We are pleased to inform you that your manuscript 'An off-lattice discrete model to characterise filamentous yeast colony morphology' has been provisionally accepted for publication in PLOS Computational Biology.

Best regards,

Attila Csikász-Nagy

Academic Editor

PLOS Computational Biology

Daniel Beard

Section Editor

PLOS Computational Biology

Feilim Mac Gabhann

Editor-in-Chief

PLOS Computational Biology

Jason Papin

Editor-in-Chief

PLOS Computational Biology

---

## [Editor Report · Acceptance letter]

14 Nov 2024

PCOMPBIOL-D-24-01245R2 

An off-lattice discrete model to characterise filamentous yeast colony morphology

Dear Dr Jiranek,

I am pleased to inform you that your manuscript has been formally accepted for publication in PLOS Computational Biology. Your manuscript is now with our production department and you will be notified of the publication date in due course.

With kind regards,

Anita Estes
